# The Vector Competence of Asian Longhorned Ticks in Langat Virus Transmission

**DOI:** 10.3390/v16020304

**Published:** 2024-02-16

**Authors:** Yan Xu, Jingwen Wang

**Affiliations:** Ministry of Education Key Laboratory of Contemporary Anthropology, School of Life Sciences, Fudan University, Shanghai 200438, China; 19110700007@fudan.edu.cn

**Keywords:** Langat virus, *Haemaphysalis longicornis*, transstadial transmission, horizontal transmission

## Abstract

*Haemaphysalis longicornis* (the longhorned tick), the predominant tick species in China, serves as a vector for a variety of pathogens, and is capable of transmitting the tick-borne encephalitis virus (TBEV), the causative agent of tick-borne encephalitis. However, it is unclear how these ticks transmit TBEV. Langat virus (LGTV), which has a reduced pathogenicity in humans, has been used as a surrogate for TBEV. In this study, we aimed to investigate the vector competence of *H. longicornis* to transmit LGTV and demonstrate the efficient acquisition and transmission of LGTV between this tick species and mice. LGTV localization was detected in several tick tissues, such as the midgut, salivary glands, and synganglion, using quantitative PCR and immunohistochemical staining with a polyclonal antibody targeting the LGTV envelope protein. We demonstrated the horizontal transmission of LGTV to different developmental stages within the same generation but did not see evidence of vertical transmission. It was interesting to note that we observed mice acting as a bridge, facilitating the transmission of LGTV to neighboring naïve ticks during blood feeding. In conclusion, the virus–vector–host model employed in this study provides valuable insights into the replication and transmission of LGTV throughout its life cycle.

## 1. Introduction

Ticks are important hematophagous arthropods and transmit a variety of pathogens, including viruses, bacteria, and parasites. In recent years, the emergence and re-emergence of infectious tick-borne viral diseases have posed considerable challenges to both animal and human health, as well as social economics [1,2]. For instance, severe fever with thrombocytopenia syndrome virus (SFTSV) was initially identified in rural China in 2009, and later reported in Japan and Korea in 2012 [3,4]. The SFTS disease can result in a severe hemorrhagic fever, leukopenia, encephalitis, and multiple organ failure in severe cases, with a mortality rate of 12–50% [5]. Tick-borne encephalitis (TBE), caused by the tick-borne encephalitis virus (TBEV), a tick-borne flavivirus (TBFV), is prevalent in Europe and Asia, with around 11,000 human cases reported per year, mostly in Russia [6,7]. The morbidity and mortality of TBE depends on the viral subtype. Although infection with any subtype is serious, Far Eastern TBEV (TBEV-FE) infection is particularly severe, with a mortality rate of 40% [8]. Other TBFVs include Powassan virus (POWV), Omsk hemorrhagic fever virus (OHFV), and Kyasanur Forest disease virus (KFDV), which also pose threats on human health [9]. For most tick-borne viral diseases, there are currently no specific pharmacological interventions available [7,10]. Therefore, it is crucial to enhance our understanding of viral infection and transmission in both the hosts and tick vectors. However, research on this topic is limited due to the requirement of high-security facilities for most tick-borne viruses.

Langat virus (LGTV), a TBFV, is considered a nonpathogenic or low-pathogenic tick-borne flavivirus for humans. This naturally attenuated LGTV is genetically and antigenically closely related to TBEV, making it a suitable biosafety level two model for TBEV infections [11,12]. Consequently, LGTV has been employed as a surrogate for TBEV in research. While previous studies on LGTV have primarily relied on mouse models [12,13] and mammalian cell lines [11,14,15,16,17], the mechanisms governing LGTV maintenance and transmission in ticks have remained elusive.

In this study, *H. longicornis* serve as a competent vector for LGTV. Once invading midgut cells, LGTV quickly disseminates to salivary glands and the synganglion. Notably, LGTV is sustained in various developmental stages through transstadial transmission, although we do not see any evidence of vertical transmission. Furthermore, LGTV exhibits the ability to transmit between mice and ticks reciprocally through blood feeding. Importantly, LGTV can spread to the adjacent uninfected ticks during co-feeding on the same mouse, suggesting that *H. longicornis* could also serve as a reservoir for this virus. Altogether, these findings offer valuable insights into transmission within ticks and may shed light on the replication and transmission mechanisms of other tick-borne flaviviruses in tick vectors.

## 2. Materials and Methods

### 2.1. Tick Maintenance

*H. longicornis* (parthenogenesis strain) originally from Yunnan Province, China, were maintained in the insectary at Fudan University at 25 °C with 85% relative humidity and a 12 h/12 h light/dark photoperiod. Each stage of *H. longicornis* was sustained by feeding them on eight-week-old male *BALB/C* mice (purchased from JSJ in Shanghai, China) [18].

### 2.2. Cell Culture and Virus Amplification

BHK-21 cells (a gift from professor Zhenhua Zheng, Wuhan Institute of Virology, Chinese Academy of Sciences, Wuhan, China) were cultured in Dulbecco’s modified Eagle’s medium (DMEM) supplemented with 10% fetal bovine serum (FBS). The cells were maintained at 37 °C in a 5% CO_2_ incubator [19]. The LGTV TP21 strain utilized in this study was also kindly provided by professor Zhenhua Zheng. Virus amplification was carried out in BHK-21 cells via infecting them at a multiplicity of infection (MOI) of 0.1. Supernatants were collected when cell mortality reached 70% [14,20]. The virus stock was aliquoted and stored at −80 °C.

### 2.3. Plaque Assays

The LGTV titer was determined using the standard plaque assay on BHK-21 cells. Adult ticks were homogenized in PBS containing 1% penicillin-streptomycin (PS), and the supernatant was used for the plaque assay. Briefly, BHK-21 cells were seeded in a 6-well plate at a density of 1.5 × 10^5^ cells/mL and incubated at 37 °C overnight. Once the cells reached 90% confluence, they were washed with PBS, and a serial dilution of tick supernatant or viruses was added to infect the cells. After a 2 h incubation at 37 °C, unabsorbed viruses were removed via washing with PBS. The infected cells were overlaid with 0.5% agarose in 2 × MEM containing 2% FBS, followed by incubation at 37 °C for 4 days. After this incubation period, the cells were fixed with 4% paraformaldehyde (PFA) at 4 °C overnight. Then, the agarose was removed and the fixed cells were stained with Coomassie brilliant blue. The virus titer was calculated as plaque forming units (PFU)/mL [21].

### 2.4. Tick Infection

Ticks were infected with LGTV through two microinjections methods, hemolymph microinjection and anal pore microinjection [18,19], as well as via feeding on LGTV-infected mice. For microinjection, 15 nL or 400 nL 1 × 10^7^ pfu/mL LGTV was injected into nymph and adult ticks, respectively. Equal volumes of DMEM medium-injected ticks were used as controls. After injection, ticks were placed in a breathable jar with wet tissue [22]. To acquire LGTV through blood feeding, approximately two-month-old larvae and nymphs were allowed to feed on LGTV-infected mice until fully engorged.

### 2.5. Mice Infection

C57BL/6 mice deficient in the type I interferon (IFN) receptor (A6 mice), kindly provided by professor Yang Li (Institute of Zoology, Chinese Academy of Sciences, Beijing, China), were maintained in the SPF level of Fudan University. Six-to-twelve-week-old A6 mice were used for virus transmission assays. A6 mice were intraperitoneally (i.p.) administered with 100 uL of virus suspension containing 10 pfu LGTV [23]. A6 mice that received an equivalent volume of DMEM were used as controls. Nymphs were allowed to bite infected A6 mice 3 hpi. All animal experiments were carried out in accordance with the guidelines of the Experimental Animal Welfare and Ethics Committee of Fudan University.

### 2.6. RNA Extraction and Quantitative Real-Time PCR (qRT-PCR)

For RNA extraction, whole ticks and various tissues, including midguts, salivary glands, synganglion, ovaries, and A6 mouse blood, were collected at indicated time points. Three nymphs or tissues, such as three salivary glands, midgut, synganglion, and ovaries, and fifty larvae or one adult tick were pooled for one biological sample, respectively [18]. RNA was isolated using the Trizol Reagent according to the manufacturer’s instruction (Yeasen). cDNA reverse transcription was performed using Hifair^®^ III 1st Strand cDNA Synthesis SuperMix for qPCR according to the protocol (Yeasen). LGTV membrane-associated glycoprotein precursor (*Pre-M*) gene-specific primers were used to detect LGTV RNA, as described in [19]. The quantification of *preM* gene expression was conducted using a standard curve. The *H. longicornis* actin protein gene-specific primers were used for normalization.

### 2.7. Transstadial Transmission Analysis of LGTV

Nymphs that had obtained the virus from viremic A6 mice at the larvae stage were allowed to feed on A6 mice. The fully engorged ticks were collected and molted to adult ticks. LGTV RNA was quantified in adults using qPCR, as described above.

### 2.8. Transovarial Transmission Analysis of LGTV

The two-month-old adult ticks were injected with LGTV through the anal pore and were allowed to feed on an *BALB/C* mice 24 h post injection. Fully engorged ticks were collected and allowed to lay eggs. LGTV presence in larvae was detected using qPCR.

### 2.9. Horizontal Transmission Analysis of LGTV

Two groups of nymphs were pooled together and allowed to feed on the same A6 mice. One group contained 15 two-month-old nymphs injected with 15 nL of 10^7^ pfu/mL LGTV via the anal pore. The other group included an equal number of DMEM-injected nymphs that were free of LGTV. After engorgement, the number of virus-positive ticks was examined using qPCR, and the infection prevalence was calculated.

### 2.10. LGTV Transmission Assay between Ticks and Mice

A6 mice infected with 10 pfu LGTV were used to infect nymphs. The fully engorged nymphs were maintained for approximately 14 days until they molted to adults. It would take 28 days for adults to become hungry and ready to take a blood meal (Day 28). The infection status of 28-day-old adults was examined using QPCR and the plaque assay. The remaining infected adult ticks were allowed to feed on A6 mice (one tick per mouse) for three days, and approximately 100 uL blood samples were collected and used for LGTV quantification.

### 2.11. Antibodies and Western Blot

The LGTV Envelope domain EDIII gene was amplified from nymphs’ cDNA and cloned into the pET-28a(+) expression vector [24]. A rabbit polyclonal anti-EDIII antibody was generated against recombinant EDIII protein (rec EDIII) in GL Biochem, Shanghai, China [25]. To determine the specificity of the anti-LGTV antibody, E protein levels were compared between ticks and cells infected with or without LGTV. Mouse anti-β-Actin antibody (1:2000) (Abbkine, Wuhan, China) was used as an internal control.

### 2.12. Immunohistochemistry

Midguts, synganglion, salivary glands, and ovaries were dissected from adult ticks. Tick organs were fixed in 4% PFA and dehydrated. Then, 5 μm thick serial sections were prepared. After deparaffinization and rehydration, slices were blocked with 3% BSA in PBST for 2 h at room temperature, followed by incubation with primary antibodies against LGTV (rabbit polyclonal LGTV-EDIII, 1:300) in 3% BSA at 4 °C overnight. The secondary antibody (Alexa Fluor 546 F(ab’)2 fragment of goat anti-rabbit IgG) was incubated for 1 h at room temperature in the dark. After washing with PBS, the slides were incubated with DAPI for 10 min. Images of sections were obtained using a Nikon positive laser scanning confocal microscope (Nikon, Duesseldorf, Germany) [18].

### 2.13. Statistical Analysis

All statistical analyses of RT-qPCR data were performed using GraphPad Prism 8 statistical software, with an unpaired Student’s *t* test with *p* < 0.05 considered statistically significant. All experiments were conducted at least three times.

## 3. Results

### 3.1. Replication of LGTV in H. longicornis

To determine the capacity of *H. longicornis* to transmit LGTV, we employed two different microinjection methods: anal pore and hemolymph injection, with 150 pfu LGTV administered to nymphs [26]. LGTV was amplified in baby hamster kidney (BHK21) cells, and titrated using the plaque assay (Figure 1A,B). The viral load was detected in nymphs on days 1, 14, 21, 28, 35, and 42 post injection (dpi). We used the expression level of the LGTV membrane-associated glycoprotein precursor (*preM)* as an indicator of virus load. Notably, we observed the successful LGTV replication in nymphs using both injection methods (Figure 1C,D). A significant proliferation was detected at 14 dpi and maintained at a relatively high level until 42 dpi (Figure 1C,D). Importantly, no significant mortalities were observed in LGTV-infected nymphs.

To stimulate the natural virus acquisition process in ticks, we infected nymphs through blood feeding on LGTV infected mice. Interferon alpha/beta receptor 1-deficient mice (*Ifnar1^−/−^*, A6 mice), which lack type I interferon responses, were used [23]. Six-to twelve-week-old A6 mice were intraperitoneally injected with 10 pfu LGTV, and nymphs were allowed to feed on these mice 3 h post injection (hpi). We collected ticks that had been feeding for two days (F2D) when the ticks were in a slow blood feeding phase, and four days (fully engorged) (F4D) (Figure 1E). Remarkably, ticks in both time points exhibited a significantly increasing *preM* expression, indicating a 100% infection prevalence of LGTV (Figure 1F,G). These findings strongly suggest that *H. longicornis* are a competent vector for LGTV.

### 3.2. The Tissue Tropism of LGTV in H. longicornis

We next investigated the tissue-specific distribution of LGTV in nymphs and adults using qPCR. The midgut, salivary glands, synganglion, and ovaries were collected from LGTV-infected nymphs and adults, respectively. Significantly higher *preM* levels were detected in the midgut, salivary glands, and synganglion of ticks at both developmental stages, and the ovaries of adults, compared to controls (Figure 2A–G). To confirm the localization of the virus in these tissues, we detected LGTV in the midgut, salivary glands, synganglion, and ovaries of adults using immunofluorescent staining. A polyclonal antibody against the domain III of envelope protein (E) was generated. The specificity of the anti-E antibody was validated in cells and ticks (Appendix A). Fluorescent signals were detected in the infected midgut, salivary glands, and synganglion of LGTV-infected adult ticks, but not in the ovaries (Figure 2H). The inconsistent results for LGTV in ovaries revealed using qPCR and immunofluorescent staining might be due to the low virus titer in the ovaries. Collectively, these results demonstrate a successful LGTV spread in various tissues of *H. longicornis*.

### 3.3. Transstadial and Transovarial Transmission of LGTV in H. longicornis

Transstadial and transovarial transmission may be crucial for the maintenance of the tick-borne flaviviruses in nature. To determine whether LGTV is maintained in *H. longicornis* through transstadial transmission, naïve larvae and nymphs were fed on LGTV-infected A6 mice until repletion (Figure 3A). It takes around 14 days for larvae and nymphs to develop into nymphs and adults, respectively. Subsequently, we detected LGTV viral load in these newly molted ticks. LGTV was transmitted to the next tick developmental stage with a 100% success rate (Figure 3B,C).

As LGTV was detected in ovaries with low *preM* RNA levels using qPCR but not via immunofluorescence, we next examined the possibility of LGTV vertical transmission to the next generation. Adult ticks were infected with LGTV via anal pore microinjection, and their larvae were analyzed for LGTV (Figure 3D). We randomly collected six adults on one dpi and examined their infection status. All adults were successfully infected with LGTV, as expected (Figure 3E). However, when the remaining infected adults were allowed to feed on *BALB/C* mice individually (one adult per mice), the newly molted larvae from these infected mothers showed low detectable viral RNA levels (Figure 3F), no difference compared to the control group. These results suggest that LGTV is maintained in ticks at different developmental stages within the same generation, but did not suggest evidence of spreading across generations.

### 3.4. Transmission of LGTV from H. longicornis to Mice

Given that LGTV can invade tick salivary glands, we next investigated whether this virus was transmitted from ticks to mice during blood feeding. We first examined the infection prevalence of adult ticks that acquired LGTV during their nymphal stage through randomly selecting adults, all of which tested positive for LGTV via qPCR and virus plaque assays (Figure 4A,B). Starved adults were allowed to bite A6 mice in a one-tick-per-mouse manner, and on the third day post-bite, we detected the viral RNA in mouse blood, with five (m3, m10, m17, m18, and m22) of 23 mice showing viremia (Figure 4C). Here we used viremia as the indicator of transmission success in mice, but it is worth noting that in some cases, tick-borne viruses can be transmitted effectively even when the virus is not detectable in the blood [27]. In summary, our results suggest that LGTV can be transmitted to mice through tick biting.

### 3.5. Horizontal Transmission of LGTV among Ticks during Blood Feeding

*H. longicornis* not only serve as vectors but also as reservoirs for various pathogens, including SFTSV, Heartland virus (HRTV), and POWV [28,29]. To investigate whether infected nymphs can transmit the virus to naïve nymphs during blood feeding, we allowed 15 LGTV-infected and 15 uninfected nymphs to feed on the same mice until repletion and the ticks were allowed to molt to adults (Figure 5A). Viremia in mice, fully engorged nymphs, and newly molted adults were determined using qPCR. The mice developed high-titer viremia after being bitten by nymphs (Figure 5B). From the two independent experiments, we found that 78.6% and 96.7% of the total 28 and 30 nymphs were positive for LGTV after co-feeding. After molting, 83.3% and 80.8% of the total 24 and 26 ticks remained infected (Table 1). These results indicate that mice serve as a bridge for LGTV transmission from infected ticks to naïve ones, and infected ticks not only act as vectors, but also function as reservoirs for LGTV.

## 4. Discussion

*H. longicornis* are a competent vector in China. They transmit several pathogens including SFTSV, POWV, *Rickettsia*, and *Borrelia* [30]. LGTV has served as a surrogate for TBEV due to its low pathogenicity. In this study, we established a virus–vector–host transmission model and demonstrated that *H. longicornis* can be a competent vector of LGTV. Our research revealed LGTV’s infection of multiple tick organs, including midgut, salivary glands, and synganglion. While LGTV can be horizontally transmitted between mice and ticks, no evidence showed successful vertical transmission. Additionally, we identified co-feeding on the same animal as another route for LGTV to spread within the tick population.

Developing a suitable “vector–host” model is crucial for studying the acquisition and transmission dynamics of viruses. Arboviral acquisition represents an essential stage in the viral lifecycle. Our findings indicate the successful acquisition of the virus by ticks during blood feeding. In addition, the presence of detectable viral RNA in F2D ticks suggests an early and quick colonization of the virus in ticks. Our results align with previous research demonstrating *H. longicornis’* ability to transmit LGTV to susceptible mice [19]. In addition to *H. longicornis*, *Ixodes scapularis* can acquire LGTV through biting infected mice [31], highlighting LGTV’s capacity to spread through multiple tick species in nature.

The transmission of pathogens from an infected tick to hosts must overcome various tissue barriers, such as the midgut, salivary glands, and ovaries [32]. The midgut serves as the initial site of pathogen contact [33]. After invading the midguts, the pathogen migrates to the salivary glands via the hemolymph, a critical step for successful transmission during blood feeding [34]. Our research demonstrates the presence of LGTV in the midguts and salivary glands of nymphs and adults. In contrast, other tick-borne bacteria, including *Borrelia* and *Bartonella* species, mainly reside in the midgut and do not migrate to salivary glands until blood feeding [35,36]. However, the exact timing of LGTV’s transition from the midgut to salivary glands remains unclear.

The tick’s central nervous system (CNS), known as the synganglion, plays a pivotal role in synthesizing and releasing signaling molecules that regulate tick behavior and physiology [37]. Our study indicates LGTV’s infection in the synganglion of *H. longicornis*, consistent with previous findings that LGTV^GFP^ can infect and replicate in synganglion culture ex vivo [38]. However, it remains uncertain whether synganglion infection by LGTV influences tick behavior, a question worthy of future investigation.

Transovarial transmission represents a common strategy for tick-borne pathogens to maintain their infection in nature. SFTSV, HRTV, and TBEV are all transmitted from mother to offspring [39,40]. In contrast, our study reveals that although a low level of LGTV was detected in ovaries, no evidence showed vertical transmission was successful in *H. longicornis*. This suggests that multiple factors influence the LGTV’s transmission success in ovaries.

In nature, infected and uninfected arthropod vectors usually feed on animals together. In our study, we show that infected nymphs transmit viruses to naïve nymphs though co-feeding on the same mouse, further highlighting the efficient dissemination capability of ticks. The bitten mouse also exhibits viremia. These results indicate that the infected nymphs not only serve as a vector, but also function as reservoirs for LGTV. In some cases, the infection of hosts by the virus may not play a decisive role in tick–tick horizontal transmission. For instance, uninfected ticks still acquire Thogoto virus from infected ticks through co-feeding on the same guinea pigs, even when the vertebrate host does not develop viremia [41,42]. These findings suggest complex interactions between viruses, ticks, and hosts. Understanding these interactions between ticks and viruses would provide significant opportunities to identify targets for controlling tick-borne viruses and preventing the diseases they cause.

## 5. Conclusions

Tick-borne encephalitis virus (TBEV) is the causative agent of tick-borne encephalitis, a sometime fatal neurotropic infection with no available therapy. Using the low-pathogenic surrogate Langat virus (LGTV), we demonstrate that *H. longicornis*, the most prevalent tick species in China, can be a competent vector. LGTV rapidly colonizes and disseminates to the salivary glands and synganglion of *H. longicornis* within two days of blood feeding. It is sustained transstadially, but vertical transmission was not demonstrated. Moreover, LGTV transmits horizontally to adjacent naïve ticks using the mammalian host as a bridge. These findings provide novel insights into flavivirus–tick interactions and transmission mechanisms, enhancing the understanding of TBEV pathogenesis and transmission dynamics with implications for disease control.

## Figures and Tables

**Figure 1 viruses-16-00304-f001:**
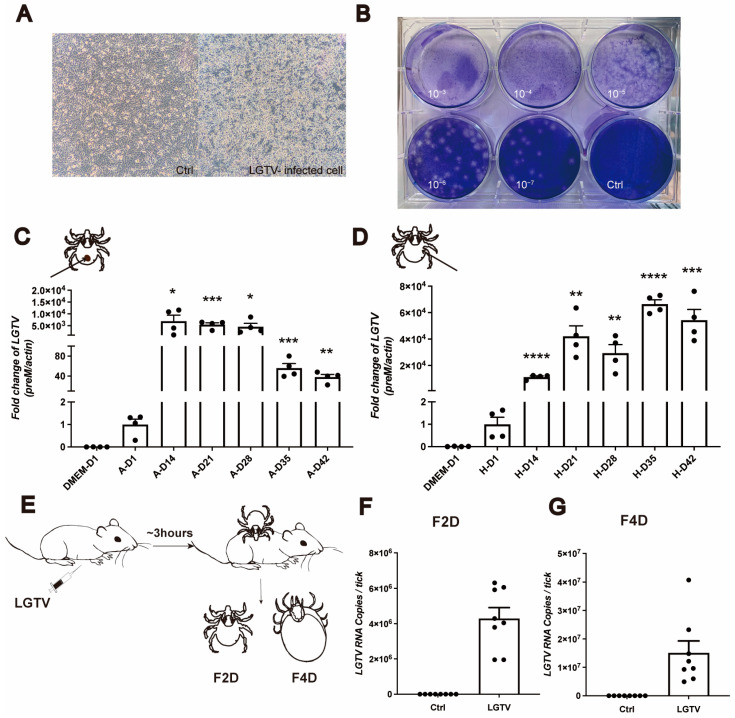
Replication of LGTV in *H. longicornis* nymphs through microinjection. (**A**) Cell image of LGTV-BHK21 cell. (**B**) The supernatant of LGTV-BHK21 cells was collected. Viral particles are serially diluted and added onto confluent cell monolayers of BHK21, and the plaque assay was performed to determine viral titers. (**C**,**D**) *H. longicornis* nymphs were injected with LGTV via anal pore (**C**) and hemolymph injection (**D**).Viral RNA was determined using RT-qPCR at the indicated time points. The expression level of *preM* was normalized to actin. The relative expression levels of *preM* in ticks from 14 days post injection (D14) to 42 days post injection (D42) were normalized to those in ticks one day post injection (D1). Three nymphs were pooled for one biological replicate. Each dot represented a biological replicate. (**E**) Schematic depiction of the experimental design. Six-to-twelve-week-old A6 mice were infected intraperitoneally with 10 pfu LGTV. The nymphs were allowed to bite the infected A6 mice. LGTV was quantified in nymphs that had been feeding for two days (**F**) or four days (**G**). Two nymphs were pooled for one biological replicate in (**F**), and an individual engorgement nymph was one biological replicate in (**G**). Each dot represents a biological replicate. Significance was determined using Student’s *t* test. * *p* < 0.05, ** *p* < 0.01, *** *p* < 0.001, **** *p* < 0.0001.

**Figure 2 viruses-16-00304-f002:**
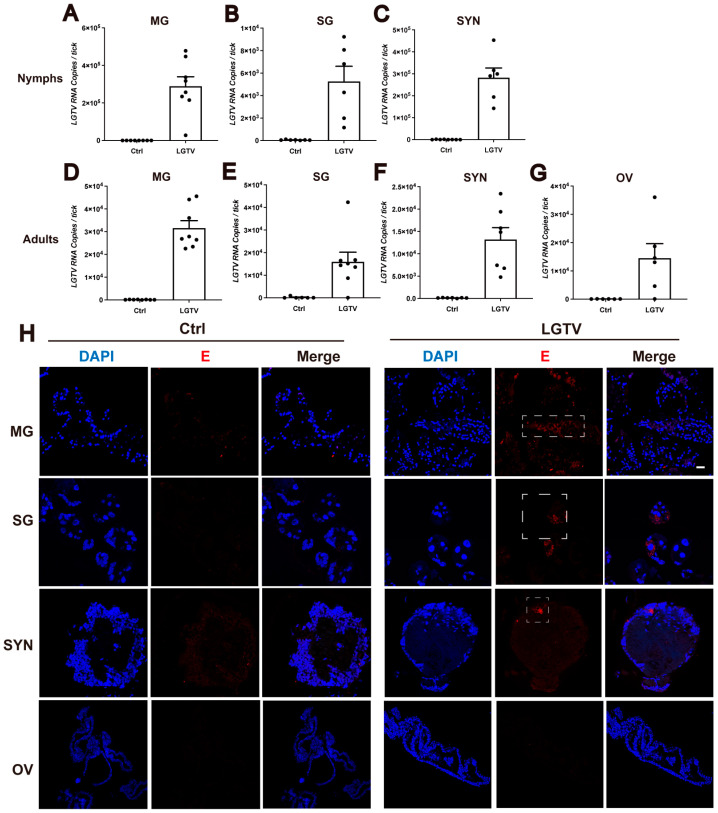
LGTV is detectable in various tick organs. (**A**–**G**) Viral RNA from midguts (**A**,**D**), salivary glands (**B**,**E**), synganglion (**C**,**F**), and ovaries (**G**) of engorgement nymphs (**A**–**C**) and adult ticks (**D**–**G**) were quantified using RT-qPCR. Three midguts, three salivary glands, three synganglion, or three ovaries were pooled for one biological replicate. Each dot represents a biological replicate. (**H**) Tissue localization of the LGTV (red) in infected ticks from adult ticks. Viral antigens were detected using a specific LGTV anti-EDⅢ polyclonal antibody, while the organ from normal ticks which bit DMEM-injected A6 mice served as a control. Nuclei counterstaining (blue) was performed using DAPI. Dashed boxes denote LGTV. Images were representative of 10 ticks. Scale bars, 25 µm. MG, midgut; SG, salivary glands; SYN, synganglion; OV, ovary.

**Figure 3 viruses-16-00304-f003:**
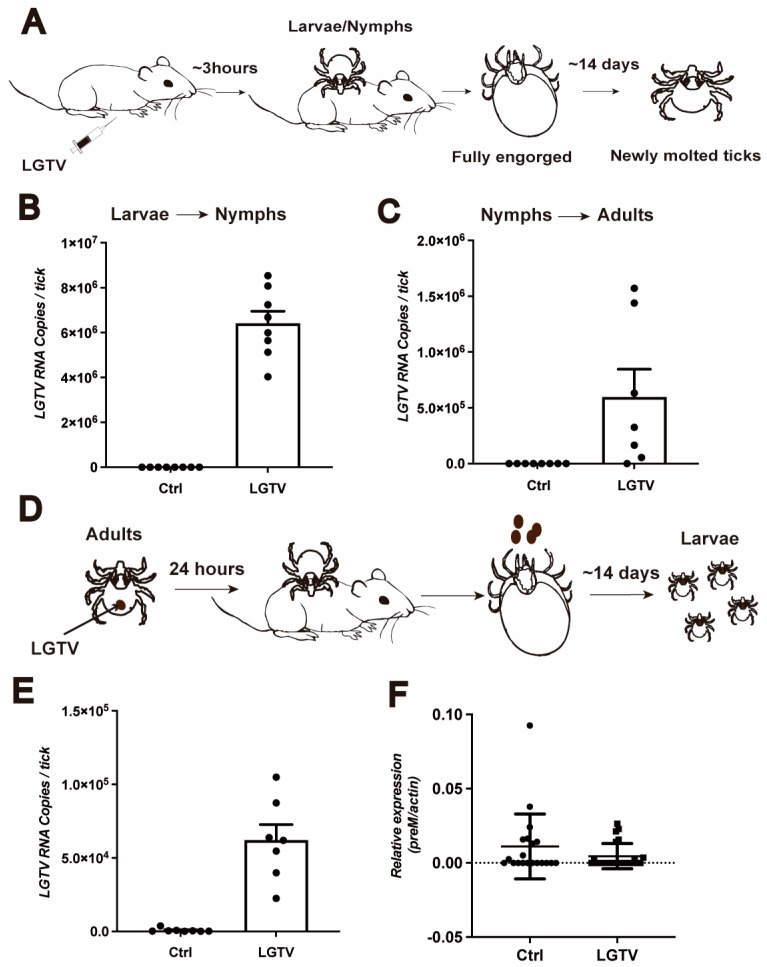
LGTV is transstadially but not transovarially transmitted in *H. longicornis*. (**A**) Schematic depiction of the experimental design. Six-to-twelve-week-old A6 mice were infected intraperitoneally with 10 pfu LGTV, and the larvae or nymphs were allowed to bite the infected A6 mice. The engorged larvae or nymphs were collected and left to molt to the next stage. Viral RNA was quantified using qRT-PCR. (**B**,**C**) The viral RNA of the next stage, nymphs or adults, was quantified. Each dot represents three nymphs or one adult tick. (**D**) Schematic representation of the experimental design. Adult ticks were injected with 4000 pfu LGTV and the infected adult ticks were allowed to bite *BALB/C* to engorgement. The engorged adult ticks laid eggs and molted to larvae. (**E**) The infection of adult ticks was detected at 1 dpi. Each dot represents an adult tick. (**F**) RT-qPCR was performed to determine the viral RNA of larvae. A total of 50 larvae were pooled for one biological replicate. Each dot represents one biological replicate.

**Figure 4 viruses-16-00304-f004:**
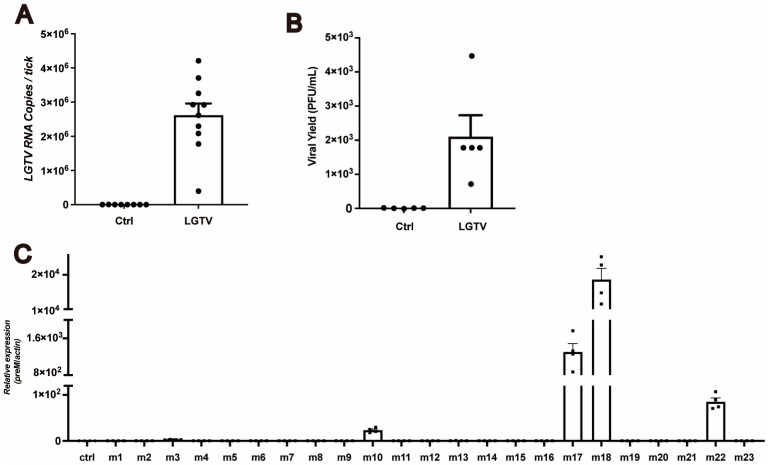
LGTV is transmitted from infected ticks to A6 mice or naïve nymphs. (**A**) The nymphs obtained LGTV from infected A6 mice. The engorged nymphs were collected (Day 0 ticks) and after 14 days had molted to adult ticks. It would take 28 days for adults to become hungry and ready to take a blood meal (Day 28). Viral RNA was detected using qRT-PCR at the appropriate time points. Each dot represents an individual tick. (**B**) The plaque assay was performed to determine viral titers within the supernatant of infected adults 28 days post molting to adults. (**C**) Transmission from LGTV-infected adult ticks to susceptible A6 mice. LGTV-infected ticks (Day 28) were allowed to bite A6 mice, one tick per A6 mouse. The viral RNA of the blood-fed-upon A6 mice was determined three days post feeding. Each dot represents a blood sample collected from one A6 mouse.

**Figure 5 viruses-16-00304-f005:**
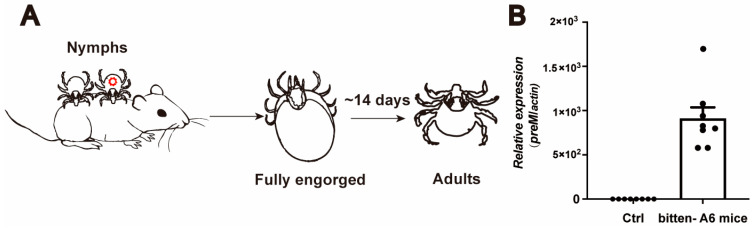
LGTV is transmitted from infected ticks to naïve nymphs through horizontal transmission. (**A**) Schematic depiction of the experimental design. A total of 15 nymphs were injected with 150 pfu LGTV via anal pore microinjection (red circle), 15 nymphs injected with 15 nL DMEM medium served as a control. A total of 30 treated nymphs were allowed to co-feed on one A6 mouse until engorgement and allowed to molt to an adult tick. Viral RNA was extracted and determined using RT-qPCR. (**B**) Blood samples were collected when the nymphs were engorged.

**Table 1 viruses-16-00304-t001:** LGTV transmission from infected adult ticks to A6 mice ^a^.

Treatment: LGTV/DMEM-Injected Nymphs	LGTV Detection	Percentage% (Positive/Total)
15 + 15 (Exp1.)	Engorged nymphs	78.6% (22/28)
15 + 15 (Exp2.)	Engorged nymphs	96.7% (29/30)
15 + 15 (Exp1.)	Adult ticks	83.3% (20/24)
15 + 15 (Exp2.)	Adult ticks	80.8% (21/26)

^a^ For the initial feeding, 15 LGTV-infected nymphs and 15 uninfected nymphs were allowed to feed on the same mouse until repletion.

## Data Availability

No new data were created.

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
