# Peer review of "The Vector Competence of Asian Longhorned Ticks in Langat Virus Transmission"

_viruses, 2024, doi:10.3390/v16020304_

Round 1
Reviewer 1 Report
Comments and Suggestions for Authors
This is a well written manuscript and the methods were well thought out. As we continue to see an increase in tick-borne illnesses it is important that we understand transmission dynamics. This paper used LGTV as a model for TBEV to better understand infection and dissemination. There were only a few grammatical and sentence structure issues that stood out. However, the manuscript should be checked to make sure that no others are present.
Comments on the Quality of English LanguageAbstract: Line 10. The sentence starting with However,…is the wrong sentence structure. Should be However, it is unclear how these ticks transmit …
Introduction Line 36-37
but Far Eastern TBEV (TBEV-FE) infection is particularly severe, with a mortality rate of 40%--is the Far eastern correct grammar?
Line 312 after invading the midgut
Check the manuscript for more grammatical and sentence structure issues.
Reviewer 2 Report
Comments and Suggestions for Authors
Xu and Wang submitted a manuscript titled “The vector competence of Asian longhorned tick in Langat virus transmission.” This is an interesting study examining the competence of H. longicornis as a host for the tick-borne flavivirus, Langat virus. While this manuscript has valuable data, there are some issues both with the presentation and the design of the study. Particularly, presenting the RT-qPCR data in relative ratios does not offer much information as the controls are always zero. A standard curve should be used to allow quantitation and expression in RNA copies. Additionally, t-tests on control vs virus infected samples seem unnecessary when the only primer being used in addition to the housekeeping gene is for a viral protein. Additionally, while cited early in the manuscript, work by Raney et al (Front Cell Infect Microbiol) on the horizontal and vertical transmission of POWV by H. longicornis, should be compared in the discussion to the author’s results. Furthermore, corrections to the English within the manuscript will help with clarity and ease of reading.
Comments on the Quality of English LanguageI have the following recommendations to the authors:
1) Line 10: “TEBV”; should read “TBEV”
2) Lines 16-18: “LGTV can be transmitted horizontally to different developmental stages within the same generation, but not vertically”; I would recommend saying “We demonstrated horizontal transmission of LGTV to different developmental stages within the same generation but did not see evidence of vertical transmission.”
As the authors saw a decrease in the preM/actin ratio from life stage to life stage, it’s possible that the amount of virus transmitted vertically was below the limit of detection. I don’t think your data rules out vertical transmission, you just didn’t see evidence of vertical transmission
3) Line 34: “ tick-borne encephalitis virus (TBEV)”; I’d add that it’s a tick-borne flavivirus: “tick-borne encephalitic virus (TBEV), a tick-borne flavivirus (TBFV), is prevalent”
4) Line 36: “TBE depend on the viral subtype”; change to “depends”
5) Line 36: “Although infection with any subtype is serious, but Far”; remove “but”
6) Line38: “Another TBEV group includes POWV”; change to “Other TBFVs include Powassan virus, etc” as POWV, OHFV, & KFDV aren’t part of the TBEV serocomplex
7) Line 39: “Kyasanur Forest disease virus (KFDV) also pose”; switch to “(KFDV) which also pose”
8) Line 44: “a member of the TBEV group”; change to “LGTV, a TBFV, is”
9) Line 51: “elucidate that H. longicornis serve as”; change to “H. longicornis can serve as”
10) Lines 53-54: I don’t believe LGTV has been shown to be incapable of vertical transmission; it would be surprising if it couldn’t be vertically transmitted as POWV in H. longicornis has shown vertical transmission (Raney et al, DOI: 10.3389/fcimb.2022.923914)
11) Line 57: “H. longicornis also serves”; change to “H. longicornis could also serve”
12) Line 68: “Wuhan institute of virology,”; change to “Wuhan Institute of Virology”
13) Line 69: “Chinese academy of sciences)”; change to “Chinese Academy of Sciences)”
14) Line 70-71: “cells were maintained in 37oC under a 5%”; change to “cells were maintained at 37oC in a 5%”
15) Line 85-86: “cells were fixed by 4% PFA at 4oC overnight. Then agarose gels were then removed”; change to “cells were fixed with 4% paraformaldehyde (PFA) at 4oC overnight. Then the agarose was removed”
16) Line 91: “For microinjection, 15nL and 400nL 1X107 pfu/ml viruses were”; change to “For microinjection, 15nL or 400nL of 1X107 pfu/ml LGTV was injected”
17) Line 93: ticks were placed in breathable jar; change to “ticks were placed in a breathable jar”
18) Line 94: approximately two-mouth old; change to “approximately two-month-old”
19) Line 112-115: PreM/actin ratios- it would be better to evaluate all of your qPCR data using a standard curve and expressing the values as RNA copies/ng RNA. Most of your qPCR compares an uninfected control cohort with an infected cohort, as the uninfected control is always negative, reporting the difference between the fold change between the group gives little information. Whereas, if you reported it in RNA copies/ng RNA it would allow for a better feel of how much the virus has replicated compared to the input amount of virus, particularly as you have different amounts of ticks included as 1 biological replicate for different life stages.
20) Line 131: A6 mice infected with around 10 pfu; remove “around”
21) Line 132: “until they molting to”; change to “until they molted to”
22) Line 148-149: “fixed in 4% paraformaldehyde and dehydration. Serially sectioned at 5 uM tick were prepared”; change to “fixed in 4% PFA and dehydrated. 5uM thick serial sections were prepared”
23) Line 152-153: “Secondary antibodies used were Alexa Fluor 546 F(ab’)2 fragment of goat anti-rabbit IgG for 1 hr at room temperature in the dark”; change to: The secondary antibody (Alexa Fluor 546 F(ab’)2 fragment of goat anti-rabbit IgG) was incubated for 1 hr at room temperature in the dark
24) Line 156-159: Unpaired Student’s t test; For Figure 1F, 1G, 2A-G, 3B-C,3E, 4A-C, & 5B I don’t think you should be reporting statistical significance. It appears you’re testing all of the infected samples against the uninfected for the t-test but you’re looking for preM so of course all of it comes up as significant as your control should never have preM.
25) Line 177-178: “Remarkably, all blood-few ticks, including F2D and F4D,”; Did you have ticks in addition to the F2D & F4D groups?
26) Line 180: change potential to “competent”
27) Line 197: Significantly higher preM levels detected in MG, SF, Syn, Ovaries- in your methods/legend you said you used 3 tissues per data point. Did you use the same amount of cDNA regardless of tissue? Do you know what your limit of detection is? Do you know that actin is present at constant levels throughout the tick tissues and life stages?
28) Line 201/Figure 2H: Immunofluorescent staining: I don’t see the red E label in these images. The labeling is far too faint. Higher quality images should be provided. Also, these images don’t seem to be high enough resolution as all the arrows are fuzzy and actually don’t really look like arrows
29) Line 220: “Transstadial and transovarial transmission are crucial for the maintenance”; would change to transmission may be crucial
30) Line 227-228: “immunohistochemistry”; would change to “immunofluorescence”
31) Line 232-233: “allowed to feed on Balb/c mice individual”; why use Balb/c for this rather than A6 mice?
32) Line 238: “LGTV is transstadially but no transovarially”; change no to “not”
33) Line 241: “engorgement larvae or nymphs were collected and when to molt to next stage. Viral RNA”; change to “engorged larvae or nymphs were collected and left to molt to next stage. Viral RNA”
34) Line 244: 4000 pfu? Is this a typo
35) Line 245: “ The engorgement”; change to “The engorged”
36) Lines 256: “one-tick per mice”; change to “one-tick per mouse”
37) Line 264-266: “The engorgement nymphs are collected, named Day0 ticks, after 14 days, the engorgement nymphs were molted to adult ticks (Day14). Day 42 was hungry adult ticks.”; change to “The engorged nymphs were collected (Day 0 ticks) and, after 14 days had molted to adult ticks (Day 14). By Day 42, the adults were hungry and ready to take a blood meal”
38) Line 267/Figure 4B: Plaque assay was performed to determine viral titers within the supernatant; What time point was this plaque assay performed? Are these supernatants from the exact Day 42 adults that were fed on the mice in Fig 4C?
39) Line 269-271/Figure 4C: “one tick bit one A6 mice, viral RNA of the blood of bit A6 mouse were quantified by RT-qPCR 3 days post bitten”; change to “one tick per A6 mouse. Viral RNA of the blood of fed upon A6 mice was quantified by 3 days post feeding”
However, the viral RNA wasn’t quantified you presented only preM/actin ratio data. Did you correlate the viral yield (Figure 4B) with the RT-qPCR data of each mouse (Figure 4C)? Is this the experiment described in your methods (lines 130-136)? The methods mention adult ticks 28 dpi feeding on mice and blood samples used for LGTV quantification, but your legend doesn’t mention Day 28.
40) Lines 278-279/Figure 5B: “We quantified viremia in mice, fully engorged nymphs and newly molted adults using qPCR”; The viremia wasn’t quantified it was shown in relative preM/actin ratios. Also, the qPCR data for the engorged nymphs and molted adults should be shown on this graph as well
41) Lines 280-282: Table 1; I don’t see Table 1 anywhere
42) Line 287: “15 nymphs were allowed to inject 150pfu”; change to “15 nymphs were injected with 150pfu”
43) Line 288: “served as control”; change to “served as a control”
44) Lines 289-290: one A6 mice, when to engorgement or molt to adult ticks. Viral RNA”; change to one A6 mouse to engorgement and allowed to molt to an adult tick. Viral RNA”
45) Line 297: “H. longicornis is a competent vector”; change to “H. longicornis can be a competent vector”
46) Line 298: “revealed LGTV’s infection on multiple tick organs”; change to “revealed LGTV’s infection of multiple tick organs”
47) Line 303: “Arboviral acquisition represents is an essential”; change to “Arboviral acquisition represents an essential”
48) Line 308: “Ixodes scapularis can acquits LGTV”; change to “Ixodes scapularis can acquire LGTV”
49) Line 312: “after invading midguts, the virus migrates”; change to “after invading midguts, the pathogen migrates”
50) Line 335: “viremic”; change to “viremia”
51) Line 340: “viremic”; change to “viremia”
52) Line 346: “a fatal neurotropic”; change to “a sometimes fatal neurotropic”
53) Line 348: “in China, is a competent vector”; change to “in China, can be a competent vector”
54) Line 350: “not transmitted vertically”; change to “vertical transmission was not demonstrated.”
55) Line 382 & 383: these two references need to be properly cited
56) Supplemental: There was no figure legend included for this figure. I see mention of the immunofluorescence from S1B mentioned on line 203 but don’t see any of the other figures referenced in the manuscript. While I’m not sure what the S1C western blots are showing due to the lack of reference in the text or a figure legend, it’s surprising that there’s such difference between the actin levels between the two blots, particularly when the E band is so much darker in the Day21 blot, but the actin is dramatically fainter. It makes me again wonder about using preM/actin ratios for the RT-qPCR
Round 2
Reviewer 2 Report
Comments and Suggestions for Authors
· Line 37-38 “Far Eastern TBEV (TBEV-FE) infection is particularly severe, which with a mortality rate of 40%”: Should read “Far Eastern TBEV (TBEV-FE) infection is particularly severe with a mortality rate of 40%”
· Figures:
o I see the qPCR data in the revised has changed but I’m not entirely clear how. I appreciate that the methods were changed to reflect the use of a standard curve (lines 115-116) and that the authors indicated in their response level that of cDNA used was consistent between the same types of tissues. However, in the figures some of the images changed and others did not. For example, Fig 1C & D still list preM/actin as the y-axis label and the scale and all the data points are unchanged, however, 1 F & G appear to have been changed. In 1F & G, the y-axis label is now listed as preM ng/actin ng which means it’s still a ratio but the scale has changed dramatically. The data points are also quite different in the revised version. Revised Figure 2A-G have different scales and some data points have changed. This continues with the remaining figures. These same inconsistencies exist in the other figures
I’m not sure exactly what was changed between the original and revised figures. I had suggested a standard curve be used for the RT-qPCR data (use 10 fold dilutions of a known titer LGTV in uninfected tick homogenates, plot the data, and use this curve to calculate how much virus is actually in your samples)
I believe all figures with qPCR data (Figures 1C, D, F & G; 2A-G; 3B, C, E, & F; 4A & C; 5B; S1C) should be evaluated using a standard curve to determine the actual number of RNA copies in each sample and that data should then be expressed as RNA copies/number of ticks used for the biological replicate or ng tissue used.
o #24 line 156-159: On Fig 5B, the asterisks should be removed to reflect the removal of the statistical analyses
o Table 1: Under the column “LGTV detection” I would change “Engorgment nymphs” to “Engorged nymphs”
Comments on the Quality of English Language
Two language related changes referenced above
Round 3
Reviewer 2 Report
Comments and Suggestions for Authors
The revised figures look good. I'm satisfied with the responses to all my suggestions and recommend accepting the manuscript.